# Cryoprotective Effects of Ergothioneine and Isoespintanol on Canine Semen

**DOI:** 10.3390/ani11102757

**Published:** 2021-09-22

**Authors:** Alexandra Usuga, Irene Tejera, Jorge Gómez, Oliver Restrepo, Benjamín Rojano, Giovanni Restrepo

**Affiliations:** 1Faculty of Veterinary Medicine and Animal Science, Universidad CES, Medellín 050022, Colombia; ausuga@ces.edu.co (A.U.); tejera.irene@uces.edu.co (I.T.); 2Faculty of Agricultural Sciences, Politécnico Colombiano Jaime Isaza Cadavid, Medellín 050021, Colombia; jegomez@elpoli.edu.co; 3Nutri-Solla Research Group, Solla S.A., Itagüí 423380, Colombia; orestrepor@solla.com; 4Faculty of Science, Universidad Nacional de Colombia, Medellín 050034, Colombia; brojano@unal.edu.co; 5Faculty of Agricultural Sciences, Universidad Nacional de Colombia, Medellín 050034, Colombia

**Keywords:** antioxidant, cryopreservation, dogs, reactive oxygen species, sperm

## Abstract

**Simple Summary:**

Cryopreserving dog semen allows the long-term availability of male gametes for future artificial insemination and other assisted reproductive techniques. However, freezing causes irreversible damage to sperm that can affect its ability to fertilize and generate a viable pregnancy. Sperm alterations are partly attributed to oxidation produced by reactive oxygen species (ROS); therefore, antioxidants have been included as extenders for seminal cryopreservation. The unconventional natural antioxidants might reduce deleterious changes in cryopreserved dog sperm; therefore, we evaluated the effects of cryopreservation with the antioxidants ergothioneine and isoespintanol on thawed canine sperm. Various concentrations of both antioxidants improved the movement capacity and structure of thawed spermatozoa, possibly by reducing ROS production. The unconventional antioxidants isoespintanol and ergothioneine improved the quality of cryopreserved canine semen and hence improved assisted canine reproduction.

**Abstract:**

Sperm undergo oxidative stress due to excessive production of reactive oxygen species (ROS) during cryopreservation. Some unconventional natural antioxidants can reduce ROS-induced changes in cryopreserved canine sperm. This study aimed to identify the cryoprotective effects of ergothioneine and isoespintanol on the quality of thawed canine semen. Twelve ejaculates from six dogs were cryopreserved in a tris-yolk extender without (control) or with 50 (E50), 100 (E100), or 150 (E150) µM ergothioneine or 20 (I20), 40 (I40), or 60 (I60) µM isoespintanol. We evaluated the motility and kinetics of thawed sperm using computerized analysis; determined morphology by eosin-nigrosin staining; functional membrane integrity using hypoosmotic tests, and structural membrane and acrosome integrity; mitochondrial membrane potential by fluorescence microscopy; and ROS production by spectrophotometry. Data were statistically analyzed using mixed models and Tukey tests. E100 increased total (60.6% vs. 49.6%) and progressive (26.4% vs. 20.1%) motility, straight line velocity (41.3 vs. 35.9 µm/s), and rapid sperm (17.6% vs. 12.3%) compared with controls. However, E150 reduced the numbers of hyperactive sperm. E100, I40, and I60 reduced the abnormal morphology and ROS production, and all concentrations of both antioxidants increased acrosomal integrity. We concluded that ergothioneine and isoespintanol reduce deleterious sperm alterations and oxidative stress in thawed canine semen.

## 1. Introduction

Cryopreservation enables the long-term storage of semen, but freezing has been associated with the production of free radicals that can absorb electrons from nucleic acids, proteins, and lipids, and cause cell damage [1,2]. Canine semen has an antioxidant system that mainly comprises superoxide dismutase (SOD), glutathione peroxidase (GPx), phospholipid hydroperoxide glutathione peroxidase (PHGPx), glutathione, ergothioneine (ERG), and high levels of L-ascorbic acid in all fractions. However, the canine spermatic fraction lacks activity against lipid peroxidation and catalase compared with semen from other species [3,4]. Therefore, since the development of canine artificial insemination with frozen semen, various known antioxidants such as vitamins E and C, glutathione, and butylated hydroxytoluene (BHT) have been added to the cryopreservation extenders [5,6,7,8]. However, some antioxidants have pro-oxidant effects, depending on their concentration and neighboring molecules [9,10]. Thus, unconventional antioxidants such as curcumin, kinetin, and quercetin have attracted interest in terms of improving the quality of thawed canine sperm, increasing its antioxidant capacity, and of expressing genes that modulate reactive oxygen species in sperm [11,12,13].

Therefore, we investigated the effects of 2-mercaptohistidine trimethylbetaine (ergothioneine (ERG)) and 2-isopropyl-3,6-dimethoxy-5-methylphenol (isoespintanol (ISO)), which have high antioxidant capacity and mitigate alterations in cryopreserved semen from other animal species [14,15,16]. Ergothioneine is an unusual thio-histidine betaine amino acid and potent antioxidant that is synthesized by microbes such as actinobacteria and some fungi (including mushroom fruiting bodies), but not by animals or plants that acquire it through the diet and soil, respectively [17]. Ergothioneine improves the total motility and viability of cryopreserved ram semen and decreases lipid peroxidation in cryopreserved rooster and ram sperm [14,15]. An important advantage of ERG over other antioxidants is that it does not interfere with the important roles of ROS but acts when oxidative damage becomes excessive [18]. Isoespintanol, extracted from *Oxandra cf xylopioides* (Annonaceae) leaves, is a free radical scavenger that inhibits lipid peroxidation and strengthens antioxidant enzymatic action. Furthermore, ISO is a more effective antioxidant than its biosynthetic analogue thymol [19], as it regulates mitochondrial activity by decreasing mitochondrial calcium (Ca^2+^) uptake and attenuating permeability transition pore openings in the inner mitochondrial membrane, which is related to mitochondrial swelling and dysfunction [20]. We did not find any other studies in which ERG or ISO were added to frozen canine semen. However, we hypothesized that these antioxidants would reduce reactive oxygen species (ROS) generation in canine semen during cryopreservation. This would consequently reduce structural and functional damage to sperm cells and even modulate their mitochondrial metabolism. Thus, this study aimed to define the effects of ERG and ISO on the thaw quality of thawed canine semen.

## 2. Materials and Methods

### 2.1. Sample Collection

Labrador and Golden Retriever breeds (*n* = 6 each) aged 2–10 years were housed under similar handling conditions, fed with a commercial concentrate twice daily (morning and afternoon), and given water ad libitum. Before starting the study, the dogs were clinically examined to ensure their health status. 

We applied manual stimulation to collect 12 ejaculates from the dogs into cones adapted to 15 mL plastic tubes, then the three fractions of each ejaculate were separated to minimize the amount of included prostate fluid [21]. The first and third fractions of the ejaculate were discarded, and the second (sperm fraction) was evaluated and cryopreserved.

### 2.2. Evaluation of Raw Semen

The volumes of sperm fractions from the ejaculates were measured using graduated tubes. The sperm cells were counted in one drop of raw semen using an SDM1 photometer (Minitüb GmbH, Tiefenbach, Germany), and sperm motility was assessed using an Eclipse E200 phase contrast microscope (Nikon Inc., Tokyo, Japan), thus obtaining an average of five observation fields (400×). The structural membrane integrity (SMI) and abnormal morphology (AM) were evaluated as described below. The raw semen was placed at 37 °C in a water bath in a Tris-based extender (Tris, 2.42 g; citric acid, 1.48 g; fructose, 1.00 g; glycerol, 6.4 mL; gentamicin, 25 mg; penicillin; and 50,000 IU/100 mL of ultra-pure water) until it reached a sperm count of 25 × 10^6^/mL. The extender was simultaneously supplemented with 20% egg yolk diluted 3:1 (*v*:*v*) in ultra-pure water and centrifuged at 1600× *g* for 100 min.

### 2.3. Semen Freezing and Treatment Establishment

Each ejaculate was divided into seven equal parts and supplemented without (control) or with 50 (E50), 100 (E100), or 150 (E150) µM L-ergothioneine (ERG; Sigma-Aldrich Corp., St. Louis, MO, USA) or 20 µM (I20), 40 µM (I40), or 60 (I60) µM ISO extracted from *Oxandra cf xylopioides* leaves [22]. These concentrations were established based on previous studies [14,16]. The semen samples were placed at 5 °C for 60 min, packed in V2 Dual MRS1 0.5-mL straws (IMV Technologies, L’Aigle, France) and placed horizontally for 15 min in vapor 4 cm from the surface of liquid nitrogen. The straws were stored in a liquid nitrogen tank at −196 °C and placed in a water bath at 37 °C for 1 min to evaluate the quality of thawed semen.

### 2.4. Post-Thaw Semen Quality Evaluation

#### 2.4.1. Sperm Motility and Kinetics

Seminal motility was assessed using computer-assisted sperm analysis [23]. This consisted of an E200 phase contrast microscope (Nikon Inc., Tokyo, Japan) and a Basler Scout scA780 digital camera (SODA VISION, Woodlands, Singapore) adapted to a computer equipped with SCA motility and concentration (Microptic S.L., Barcelona, Spain) software. The setup comprised a coverslip camera with 20 × 20 mm, optics in a pH (-), drop of 7 µL, dog species, thermal plate at 37 °C, and a particle size of 12–80 µm. We analyzed total motility (TM), progressive motility (PM), rapid spermatozoa (RAP), straight line velocity (VSL), curvilinear velocity (VCL), average path velocity (VAP), amplitude of lateral head displacement (ALH), beat cross-frequency (BCF), and hyperactive sperm (HYP). At least 500 spermatozoa were evaluated in five observation fields.

#### 2.4.2. Structural Membrane Integrity (SMI)

We determined SMI using Live/Dead Kits (Molecular Probes Inc., Eugene, OR, USA) [24]. The membrane-permeable nucleic acid stain SYBR14 and impermeable propidium iodide (PI) label spermatozoa with intact and compromised membranes with green and red fluorescence, respectively. Semen (20 µL containing ~20 × 10^6^ spermatozoa/mL) was suspended in Hanks Hepes solution with 1% bovine serum albumin (BSA), then incubated at 37 °C for 8 min with 6 mM SYBR14 followed by 0.48 mM PI. We analyzed 200 spermatozoa using an HBO-E200 fluorescence microscope (Nikon Inc.) with a UV-2A filter. The proportion of spermatozoa with structural membrane integrity (% SMI) was calculated from the quantity of spermatozoa with green fluorescence.

#### 2.4.3. Abnormal Morphology (AM)

We assessed AM using the modified eosin-nigrosin test as follows [18]: One droplet each of semen and eosin-nigrosin (Sigma-Aldrich Corp., St. Louis, MO, USA) were mixed and smeared on a microscope slide, which was then placed on a warming plate at 37 °C. Subsequently, 200 spermatozoa were individually classified as morphologically normal or abnormal using an Eclipse E200 phase contrast microscope.

#### 2.4.4. Functional Integrity of Membrane (FMI)

The FMI was assessed using hypo-osmotic swelling (HOS) tests. When exposed to a hypoosmotic solution, functional spermatozoa swell to establish osmotic equilibrium, resulting in typical swelling of the tail [25]. We achieved 100 mOsmol/L by incubating 100 μL of semen with 500 μL of a hypo-osmotic sucrose solution (5.4%) at 38.5 °C for 30 min. Thereafter, 200 spermatozoa were evaluated using an Eclipse E200 phase contrast microscope. Sperm with tail swelling were considered reactive, and the proportion of sperm with FMI (% FMI) was calculated.

#### 2.4.5. Acrosomal Membrane Integrity (AI)

We evaluated AI using a fluorescein isothiocyanate-conjugated peanut agglutinin (FITC-PNA) probe (Sigma-Aldrich Corp.) [26], which specifically binds to galactosyl β-1,3 N-acetylgalactosamine in acrosomal membranes [27]. These sperm samples were smeared and fixed for 10 min with 95% ethanol, dried at room temperature, and then placed in darkness for 30 min with 25 µL of FITC-PNA (5 µg/mL) in a phosphate buffer solution (PBS). The smears were then washed with distilled water, and 200 spermatozoa were evaluated using an HBO fluorescence E200 microscope with a G-2A filter. Sperm with normal or slightly disordered acrosomes were considered intact [28].

#### 2.4.6. Mitochondria Membrane Potential (ΔΨM)

We used a fluorescence microplate reader and a cationic fluorescent probe JC-1 (Molecular Probes Inc., Waltham, MA, USA) to evaluate ΔΨM [29]. The semen samples were brought to a density of 20 × 10^6^ sperm/mL, then incubated for 20 min at 35 °C with JC-1 in DMSO (2 mM). Green (low ΔΨM) and red (high ΔΨM) fluorescence intensities were evaluated as relative fluorescence units (RFUs) using an LS 55 fluorescence microplate spectrometer (PerkinElmer Life and Analytical Sciences Inc., Waltham, MA, USA) at excitation/emission wavelengths of 514/529 and 585/590 nm, respectively. Mitochondrial activity is also expressed as the ratio between high- and low-ΔΨM (red:green).

#### 2.4.7. Reactive Oxygen Species (ROS) Production

The semen samples (30 μL), buffer solution (240 μL; pH 7.4), and 30 μL of 40 mM 2,7-dichlorodihydrofluorescein diacetate (H2DCFDA) (Intervet International BV, Boxmeer, The Netherlands) were mixed to produce ROS at 37 °C [30]. The antioxidant Trolox^®^ (Merck, Darmstadt, Germany) was the reference. The fluorescence intensity was assessed every minute for 80 min in quadruplicate samples using an LS 55 spectrofluorometer (Perkin Elmer Life and Analytical Sciences Inc., Waltham, MA, USA). The results are shown as relative fluorescence units (RFUs) [31]. 

### 2.5. Statistical Analysis

We used a completely random model. To evaluate sources of variation, a mixed model was adjusted for each dependent variable (seminal quality parameters). The normality of the data was assessed using Shapiro–Wilk tests. The means were compared using Tukey tests. The level of significance for all assessments was *p* < 0.05. All data were analyzed using SAS version 9.2 software (SAS Inst. Inc., Cary, NC, USA).

## 3. Results

### 3.1. Raw Semen Quality

Table 1 shows the quality parameters of the raw canine semen. Volume, concentration, and abnormal sperm morphology were the most variable (CV > 40%).

### 3.2. Quality of Thawed Semen

We froze 245 semen straws (35 straws per ISO and ERG concentration and control) from 12 ejaculates. Table 2 shows the motility and kinetics of the thawed canine semen. The motility and kinetic parameters of sperm frozen with E100 differed from those of the control and all tested ISO concentrations. Figure 1 shows the morphology and integrity of the plasma and acrosomal membrane of thawed canine spermatozoa. The morphological alterations were reduced by E100 and the highest concentrations of ISO, and all concentrations of both agents increased the acrosomal integrity of thawed canine sperm.

The mitochondrial activity of thawed canine sperm is expressed as ΔΨM (Figure 2), from the green fluorescence of JC-1 monomers (Low-ΔΨM), the red fluorescence due to the dimerization of JC-1 (High-ΔΨM), and their relationship (red: green ratio). The mitochondrial membrane potential of cryopreserved canine semen was not affected by either antioxidant. Figure 3 shows that E100, I40, and I60 reduced ROS production in thawed canine semen.

## 4. Discussion

Seminal cryopreservation protocols entail distinct steps comprising temperature reduction, dehydration, and freezing in intra- and extracellular medium [32]. This generates increased ROS production, which can indirectly cause oxidative damage to spermatozoa [10]. Furthermore, cryopreservation significantly reduces the availability of antioxidants for cell protection, thus increasing the susceptibility of sperm cells to ROS [33]. Here, we evaluated the abilities of ERG and ISO to reduce ROS generation, oxidative stress, and the associated changes during sperm freezing. Adding ERG to the freezing extender positively affected the TM, PM, VSL, and RAP of thawed semen and, consequently, reduced deleterious changes in the morphology and integrity of acrosomes. Ergothioneine improves the movement characteristics of sperm. The supplementing of semen extenders ERG improves the PM, VAP, and the ALH of thawed ram sperm at far higher doses (2 and 4 mM) than those used herein (50, 100, and 150 µM) [34]. The ΔΨM of frozen canine spermatozoa was not modified by adding various concentrations of ERG to the extender. This agrees with the effects of even lower concentrations of this antioxidant on rooster semen [14].

We also found that ERG reduces ROS production in thawed canine sperm, indicating a mitigation effect of ERG on oxidative stress during cryopreservation. This could explain the results of other investigations in which ERG improved MT and viability, and reduced lipid peroxidation as well as DNA damage to cryopreserved ram sperm [15,35], although at much higher (2–6 mM) concentrations than those used herein. These findings imply that compared with ram semen, lower concentrations of ERG can cryoprotect canine semen. Studies of 5 and 10 mM ERG did not find any beneficial or detrimental effects on the parameters of thawed ram semen [36]. We also found that ERG did not affect sperm membrane integrity. Different concentrations of ERG negatively affect the fertility rates of thawed rooster semen [14]. Regarding the mechanism of action, ERG has potent intrinsic antioxidant activity against hydroxyl, peroxyl, and peroxynitrite radicals [37]. Ergothioneine can also react with, detoxify, or prevent the formation of singlet oxygen, superoxide, and peroxide reactive species [17].

Isoespintanol decreases mitochondrial calcium (Ca^2+^) uptake and attenuates mitochondrial permeability transition pore openings in the inner mitochondrial membrane of isolated cardiac mitochondria [16]. However, ISO did not affect the ΔΨM of sperm in the present study. Our findings also showed that ISO reduced morphological alterations, protected the acrosome integrity of thawed canine spermatozoa, and significantly reduced ROS production, which might have reduced deleterious oxidative phenomena in sperm structural components. The antioxidant activity of ISO was established using ABTS, DPPH, and TBARS assays, and by its ability to reduce Fe^3+^/TPTZ [31]. Furthermore, ISO at 40 μM can reduce ROS production in equine sperm [16]. The antioxidant effects of ISO could be attributed to those of its substituents and to the formation of intra- and intermolecular hydrogen bonds; its antioxidant role might lie in interruption of the second stage of the propagation chain of lipid oxidation by neutralizing lipid radicals (ROO^•^) [19].

The quality of cryopreserved canine semen has been evaluated [5,38]. However, evaluating the fertility of thawed canine semen in vivo is challenging, since aspects such as the conditions of the genital tract of a bitch in estrus can differently affect semen motility and viability over time [39]. In addition, other factors such as the location of thawed semen deposition, the timing of insemination, the number of inseminations, and prostatic fluid all contribute to fertility [40,41].

This is the first report to describe the effects of ergothioneine and isoespintanol on canine semen, since investigations of the effects of both antioxidants in this species have not been published as far as we can ascertain. Both antioxidants reduced morphological alterations and acrosome damage in thawed canine semen. However, ergothioneine seems to be a more effective antioxidant than isoespintanol for canine semen, as it improved the motility and kinetics and had greater capacity to reduce ROS production in thawed sperm. Future studies should evaluate the antioxidant mechanisms of ERG and ISO in canine semen, as well as their possible effects on the fertilization capacity of sperm in vitro and in vivo.

## 5. Conclusions

Ergothioneine (100 µM) and isoespintanol (60 µM) reduced deleterious changes in sperm and oxidative stress in thawed canine semen. Ergothioneine is a more effective antioxidant than isoespintanol for canine semen as it improved the motility and kinetics of thawed sperm, probably through its greater capacity to reduce ROS production.

## Figures and Tables

**Figure 1 animals-11-02757-f001:**
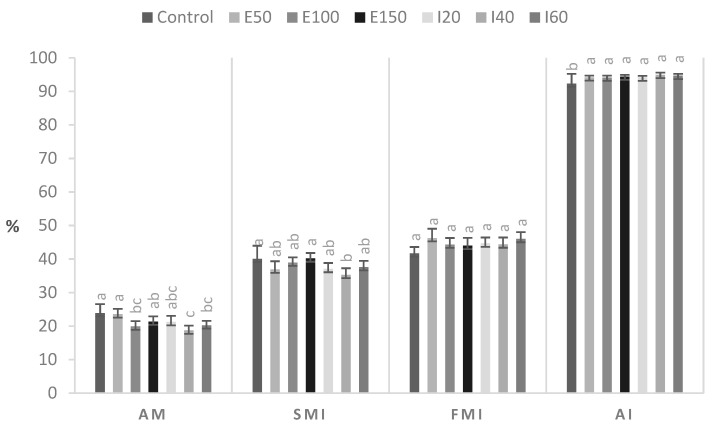
Morphology and membrane integrity of frozen-thawed canine semen. E50: ergothioneine at 50 µM. E100: ergothioneine at 100 µM. E150: ergothioneine at 150 µM. I20: isoespintanol at 20 µM. I40: isoespintanol at 40 µM. I60: isoespintanol at 60 µM. AM: abnormal morphology. SMI: Structural membrane integrity. FMI: Functional membrane integrity. AI: Acrosomal membrane integrity. Bars represent the mean ± standard error of the mean (SEM). Different letters (a, b, c) between bars indicate significant statistical difference (*p* ≤ 0.05).

**Figure 2 animals-11-02757-f002:**
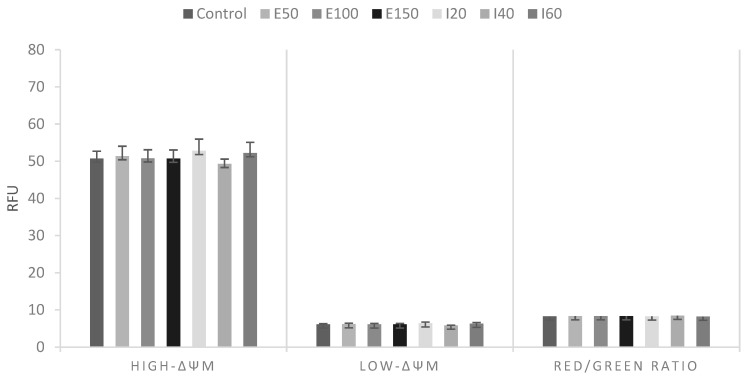
Mitochondrial membrane potential of frozen-thawed canine semen. E50: ergothioneine at 50 µM. E100: ergothioneine at 100 µM. E150: ergothioneine at 150 µM. I20: isoespintanol at 20 µM. I40: isoespintanol at 40 µM. I60: isoespintanol at 60 µM. Low-ΔΨM: low mitochondrial membrane potential. High-ΔΨM: high mitochondrial membrane potential. RFU: relative fluorescence units. Bars represent the mean ± standard error of the mean (SEM). No significant statistical differences were found (*p* ≥ 0.05).

**Figure 3 animals-11-02757-f003:**
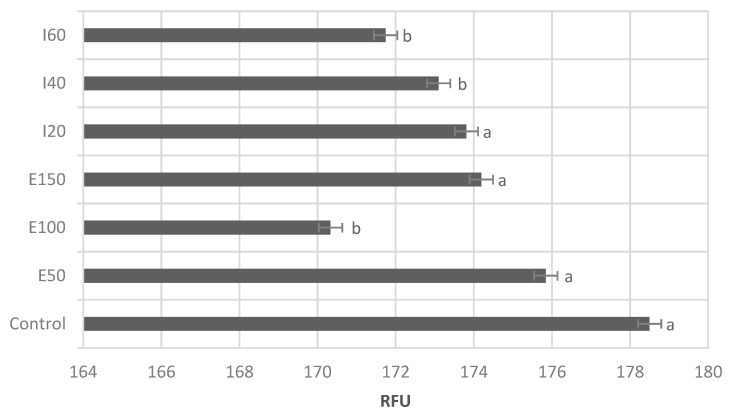
Reactive oxygen species production of frozen-thawed canine semen. E50: ergothioneine at 50 µM. E100: ergothioneine at 100 µM. E150: ergothioneine at 150 µM. I20: isoespintanol at 20 µM. I40: isoespintanol at 40 µM. I60: isoespintanol at 60 µM. RFU: relative fluorescence units. Bars represent the mean ± standard error of the mean (SEM). Different letters (a, b) between bars indicate significant statistical difference (*p* ≤ 0.05).

**Table 1 animals-11-02757-t001:** Semen parameters of raw canine semen (*n* = 12).

	Mean	SD	CV	SEM	Min	Max
Volume (mL)	1.48	0.66	44.67	0.18	0.8	3
Sperm concentration (10^6^/mL)	237.54	107.96	45.45	29.94	51	463
Motility (%)	88.46	5.55	6.27	1.54	80	95
Structural membrane integrity (%)	92.31	4.57	4.95	1.27	81	97
Abnormal morphology (%)	18.08	7.57	41.85	2.10	7	33
Functional membrane integrity (%)	88.31	6.93	7.85	1.92	76	99

SD: standard deviation. CV: coefficient of variation. SEM: standard error of the mean. Min: minimum value. Max: maximum value.

**Table 2 animals-11-02757-t002:** Motility and kinetics of frozen-thawed canine semen.

	Control	E50	E100	E150	I20	I40	I60
TM	49.61 ± 3.90 ^b^	56.61 ± 4.42 ^ab^	60.65 ± 3.56 ^a^	54.40 ± 3.63 ^ab^	54.77 ± 3.57 ^ab^	51.72 ± 3.48 ^b^	52.18 ± 4.02 ^b^
PM	20.17 ± 2.72 ^b^	24.73 ± 3.02 ^ab^	26.41 ± 2.37 ^a^	22.04 ± 2.59 ^ab^	20.17 ± 2.21 ^b^	21.41 ± 2.49 ^ab^	20.28 ± 2.84 ^b^
RAP	12.31 ± 1.91 ^b^	16.92 ± 2.24 ^ab^	17.67 ± 1.80 ^a^	14.24 ± 2.03 ^ab^	13.09 ± 1.79 ^b^	13.64 ± 1.87 ^ab^	13.60 ± 2.11 ^ab^
VCL	58.39 ± 2.88 ^ab^	60.23 ± 3.14 ^ab^	62.85 ± 2.22 ^a^	56.00 ± 2.74 ^b^	56.64 ± 2.68 ^b^	55.87 ± 2.75 ^b^	54.37 ± 3.14 ^b^
VAP	42.77 ± 2.59 ^abc^	45.50 ± 2.8 ^ab^	47.97 ± 2.08 ^a^	43.00 ± 2.38 ^abc^	41.45 ± 2.48 ^bc^	41.85 ± 2.23 ^bc^	39.26 ± 2.56 ^c^
VSL	35.97 ± 2.54 ^bc^	38.38 ± 2.88 ^ab^	41.35 ± 1.97 ^a^	37.27 ± 2.23 ^abc^	35.21 ± 2.41 ^bc^	35.77 ± 2.08 ^bc^	32.89 ± 2.41 ^c^
ALH	2.34 ± 0.09 ^a^	2.28 ± 0.11 ^a^	2.31 ± 0.07 ^a^	2.18 ± 0.08 ^a^	2.30 ± 0.09 ^a^	2.23 ± 0.08 ^a^	2.32 ± 0.12 ^a^
BCF	7.08 ± 0.40 ^ab^	7.36 ± 0.42 ^ab^	7.68 ± 0.28 ^a^	7.34 ± 0.29 ^ab^	6.96 ± 0.35 ^a^	7.01 ± 0.28 ^ab^	6.63 ± 0.39 ^b^
HYP	2.59 ± 1.12 ^a^	2.28 ± 0.84 ^ab^	2.57 ± 1.07 ^a^	1.08 ± 0.41 ^b^	1.87 ± 0.76 ^ab^	2.19 ± 0.74 ^ab^	1.42 ± 0.00 ^ab^

TM: total motility. PM: progressive motility (%). RAP: rapid spermatozoa (%). VCL: curvilinear velocity (µm/s). VAP: average path velocity (µm/s). VSL: straight line velocity (µm/s). ALH: amplitude of lateral head displacement (µm). BCF: beat cross-frequency (Hz). HYP: hyperactive spermatozoa (%). E50: ergothioneine at 50 µM. E100: ergothioneine at 100 µM. E150: ergothioneine at 150 µM. I20: isoespintanol at 20 µM. I40: isoespintanol at 40 µM. I60: isoespintanol at 60 µM. TM: total motility (%). Different letters (a, b, c) within the rows indicate significant statistical difference (*p* ≤ 0.05).

## Data Availability

No new data were created or analyzed in this study. Data sharing is not applicable to this article.

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
