# Peer review of "Cryoprotective Effects of Ergothioneine and Isoespintanol on Canine Semen"

_animals, 2021, doi:10.3390/ani11102757_

Round 1

Reviewer 1 Report

This is an interesting study and provides useful information on the effect of the addition of ergothionein and isoespintanol in canine semen freezing. The manuscript presents a well-performed experimental design, including advanced tests to examine the quality of cryopreserved semen.  According to the results in the manuscript, the addition of ergothioneine and isoespintanol reduce sperm alterations and oxidative stress of frozen-thawed canine semen. However, I have identified some minor issues that should improve the quality of the manuscript and produce a revised manuscript suitable for publication:

1- Line 105-106: Please mention that these ingredients are per 100ml or dL.

2- Was the egg yolk added after the dilution of the semen with a tris based extender or what? Please write the process of semen dilution in detail so that it is clear to the reader.

3- Line 114: Please check whether it is 1600 x g for 100 minutes or 10 minutes.

4- Line 133: Please check whether it is 200 mL or 20µL.

5- Please, write the criteria of the following tests:

  • 4.2. Structural membrane integrity.
  • 4.4. Functional integrity of membrane.
  • 4.5. Acrosomal membrane integrity.

6- Results: Please present the important results of each table and Figure.

7- Please delete:

  • Line 234: (Table 1).
  • Line 235: (Figure 1).
  • Line 241: (Figure 2).
  • Line 244: (Figure 3).
  • Line 262: (Figure 2).
  • Line 264: (Figure 1).
  • Line 265: (Figure 3).

8- Please mention the best concentration of ergothionein and isoespintanol in the conclusion.

9- Reference numbers 28 and 32 please modify the authors' names.

Best Regards

Author Response

Dear editor

The authors of the manuscript, list below the corrections made according to the suggestions issued by the reviewer # 1. We appreciate the valuable contributions of the reviewer.

Comments and Suggestions for Authors

This is an interesting study and provides useful information on the effect of the addition of ergothionein and isoespintanol in canine semen freezing. The manuscript presents a well-performed experimental design, including advanced tests to examine the quality of cryopreserved semen.  According to the results in the manuscript, the addition of ergothioneine and isoespintanol reduce sperm alterations and oxidative stress of frozen-thawed canine semen. However, I have identified some minor issues that should improve the quality of the manuscript and produce a revised manuscript suitable for publication:

1- Line 105-106: Please mention that these ingredients are per 100ml or dL.

R/ The requested description was added

2- Was the egg yolk added after the dilution of the semen with a tris based extender or what? Please write the process of semen dilution in detail so that it is clear to the reader.

R/ The description of the extender preparation and the addition of the egg yolk was improved, clarifying that the addition of the egg yolk was made from the initial preparation of the extender used for the total dilution.

3- Line 114: Please check whether it is 1600 x g for 100 minutes or 10 minutes.

R/ It was verified that the centrifugation of the yolk was carried out for 100 minutes for the appropriate separation of the fraction of the centrifuged yolk used, according to the referenced article.

4- Line 133: Please check whether it is 200 mL or 20µL.

R/ The verification and correction was carried out

5- Please, write the criteria of the following tests:

  • 4.2. Structural membrane integrity.
  • 4.4. Functional integrity of membrane.
  • 4.5. Acrosomal membrane integrity.

R/ The criteria, principles of action and classification for each parameter and probe, were included.

6- Results: Please present the important results of each table and Figure.

R/ The main results of each table and figure were presented as suggested by the reviewer.

7- Please delete:

  • Line 234: (Table 1).
  • Line 235: (Figure 1).
  • Line 241: (Figure 2).
  • Line 244: (Figure 3).
  • Line 262: (Figure 2).
  • Line 264: (Figure 1).
  • Line 265: (Figure 3).

R/ The deletions were made, as suggested by the reviewer

8- Please mention the best concentration of ergothionein and isoespintanol in the conclusion.

R/ The best concentrations for both antioxidants were mentioned in the conclusion.

9- Reference numbers 28 and 32 please modify the authors' names.

R/ References were corrected

Best Regards

Reviewer 2 Report

This is an interesting manuscript that reports the use of alternative antioxidants for canine semen cryopreservation. At general, it is well written and results bring novel information for the science

1. Tittle is adequate

2. Summary is well written and informative

3. Abstract - Despite well written, it should be a few more complete, bringing some numerical results in various ocasions. For example, authors should provide total motility both for the best group as for the others. This would help the reader to understand the extent of the improvement provided by the alternative antioxidants. Other numerical results for other parameters are also welcome.

4. Keywords are adequate.

5. Introduction - It is well written in general. However, authors should comment the advantages of ERG and ISO in comparison to other conventional or alternative antioxidants. Why should they be tested if we have a lot of other effective antioxidants? Moreover, authors should explain why they should be tested for dogs if they were previously shown efficiency for other species. What are the difference among dog sperm and other species sperm that justify the study? Do you hypothesize that the activity mechanism would be different for dogs?

6. Material and Methods are detailed and well described.

7. Results

  • Authors should revise Table 1. Tittle should be a few more informative and include details related to the number of individuals. Moreover "quality" is a term very subjective an vague. In my opinion, terms like "Semen parameters" should be more apropriate. Additionally, authors should avoid a lot of acronyms and legends. Thus, they should try to include the complete name of the parameters at the column 1. The same comments are valid for the other Tables. Please, revise all.
  • In Figure 1, the metrics for the Y-axis are absent. Please provide them.
  • In the results section, authors should include short texts summarizing the main results for each table and figure.

7. Discussions are well written and adequate in general. But authors should include discussions related to the different concentrations for both antioxidants tested. They should also discuss comparisons related to both antioxidants. What was the best one at the end?

8. Conclusions - Authors should to point the most effective concentration for both antioxidants. Moreover, they should point if one antioxidant is more effective than other or if they are similar.

Author Response

Dear editor

The authors of the manuscript, list below the corrections made according to the suggestions issued by the reviewer # 2. We appreciate the valuable contributions of the reviewer.

Comments and Suggestions for Authors

This is an interesting manuscript that reports the use of alternative antioxidants for canine semen cryopreservation. At general, it is well written and results bring novel information for the science

  1. Tittle is adequate
  2. Summary is well written and informative
  3. Abstract - Despite well written, it should be a few more complete, bringing some numerical results in various ocasions. For example, authors should provide total motility both for the best group as for the others. This would help the reader to understand the extent of the improvement provided by the alternative antioxidants. Other numerical results for other parameters are also welcome.

R/ Numerical values for total motility, progressive motility, straight line velocity and rapid sperm were included.

  1. Keywords are adequate.
  2. Introduction - It is well written in general. However, authors should comment the advantages of ERG and ISO in comparison to other conventional or alternative antioxidants. Why should they be tested if we have a lot of other effective antioxidants? Moreover, authors should explain why they should be tested for dogs if they were previously shown efficiency for other species.What are the difference among dog sperm and other species sperm that justify the study? Do you hypothesize that the activity mechanism would be different for dogs?

R/ Different references were included where advantages of ERG and ISO are presented in comparison with other antioxidants, showing the importance of evaluating these antioxidants and the disadvantages of some conventional antioxidants. Some differences in canine semen that justify the study were also cited, among which are its characteristics and limitations in terms of antioxidant components useful for mitigating damage caused by cryopreservation, for which it could be considered that the antioxidant mechanisms of canine semen and the response to ERG and ISO supplementation, may differ from that found in other species.

  1. Material and Methods are detailed and well described.
  2. Results
  • Authors should revise Table 1. Tittle should be a few more informative and include details related to the number of individuals. Moreover "quality" is a term very subjective an vague. In my opinion, terms like "Semen parameters" should be more apropriate. Additionally, authors should avoid a lot of acronyms and legends. Thus, they should try to include the complete name of the parameters at the column 1. The same comments are valid for the other Tables. Please, revise all.

R/ The title of Table 1 was modified and the number of ejaculates was included. The acronyms and legends corresponding to the semen quality parameters were eliminated and the full names were included in Table 1. The other tables and figures were reviewed, but the authors consider it inconvenient, to replace the acronyms because they are very long sentences and affect the presentation of tables and figures.

  • In Figure 1, the metrics for the Y-axis are absent. Please provide them.

R/ Metrics for Figure 1 were included

  • In the results section, authors should include short texts summarizing the main results for each table and figure.

R/ A summary of the main results for each table and figure was included.

  1. Discussions are well written and adequate in general. But authors should include discussions related to the different concentrations for both antioxidants tested. They should also discuss comparisons related to both antioxidants. What was the best one at the end?

R/ Discussions related to the different concentrations and comparisons related to both antioxidants were included. The concept regarding the best antioxidant was issued according to the results.

  1. Conclusions - Authors should to point the most effective concentration for both antioxidants. Moreover, they should point if one antioxidant is more effective than other or if they are similar.

 R/ The most effective concentration for both antioxidants and the concept of the most effective antioxidant were included in the conclusion.

Reviewer 3 Report

In the reported study, two antioxidants were added to canine semen at different concentrations before cryopreservation and post-thaw semen quality was compared. Small positive effects of ergothionein were demonstrated.

This study adds limited new information relevant for cryopreservation of canine semen. Semen freezing in dogs has been studied much less than in domestic ruminant species or horses. Results presented in the present manuscript are therefore to some extent novel but the manuscript needs to be revised before it can be recommended for publication.   

The experimental design and analytical methods to this reviewer appear to be adequate and results are not questioned. The interpretation of data should be written more careful. The authors have made multiple comparisons without adjusting the significance level. Furthermore, fertility of cryopreserved semen was not analysed. I acknowledge that such trials are seldom possible in dogs, but at least this limitation must be discussed.

The discussion would benefit from thorough language editing

Non-accessible references (not written in English) should be removed (refs 5, 12, 19, 20)

References 28 and 32 are fragments and must be replaced.

The format of references is not consistent and needs to be corrected.

Author Response

Dear editor

The authors of the manuscript, list below the corrections made according to the suggestions issued by the reviewer # 3. We appreciate the valuable contributions of the reviewer.

Comments and Suggestions for Authors

In the reported study, two antioxidants were added to canine semen at different concentrations before cryopreservation and post-thaw semen quality was compared. Small positive effects of ergothionein were demonstrated.

This study adds limited new information relevant for cryopreservation of canine semen. Semen freezing in dogs has been studied much less than in domestic ruminant species or horses. Results presented in the present manuscript are therefore to some extent novel but the manuscript needs to be revised before it can be recommended for publication.  

The experimental design and analytical methods to this reviewer appear to be adequate and results are not questioned.

The interpretation of data should be written more careful. The authors have made multiple comparisons without adjusting the significance level.

R/: Although some of the studies with which comparisons were made, used different p-values than our investigation, the authors consider that this is not a reason to eliminate the comparisons made. If the reviewer considers that the p-value of each study discussed should be explicitly stated, please indicate it, in order to make the modification. The authors do not understand what the reviewer refers to by "adjusting the level of significance" (if it is related to including the levels of significance of the cited studies in the text, or another procedure)

Furthermore, fertility of cryopreserved semen was not analysed. I acknowledge that such trials are seldom possible in dogs, but at least this limitation must be discussed.

R/: A paragraph about limitations of evaluating fertility of cryopreserved semen in dogs, was added in the discussion, as suggested by the reviewer.

The discussion would benefit from thorough language editing

R/: Language editing was performed throughout the manuscript (Certificate is attached)

Non-accessible references (not written in English) should be removed (refs 5, 12, 19, 20)

R/: References were removed according to the reviewer's suggestion. Some of them were replaced by new references.

References 28 and 32 are fragments and must be replaced.

R/: The fragment cited by reference 28 was modified and a new reference was included to complete the information. Reference 32 was not replaced due to the importance of information about the mechanism of action of ergothioneine. However, it was edited to avoid being a complete fragment.

The format of references is not consistent and needs to be corrected.

R/: The format of reference was reviewed and corrected by “Mendeley reference manager”
